# Microplastics Environmental Effect and Risk Assessment on the Aquaculture Systems from South China

**DOI:** 10.3390/ijerph18041869

**Published:** 2021-02-15

**Authors:** Yizheng Li, Guanglong Chen, Kaihang Xu, Kai Huang, Jun Wang

**Affiliations:** 1Joint Laboratory of Guangdong Province and Hong Kong Region on Marine Bioresource Conservation and Exploitation, College of Marine Sciences, South China Agricultural University, Guangzhou 510641, China; lyz0006@stu.scau.edu.cn (Y.L.); glchen@scau.eddu.cn (G.C.); xkh9203@163.com (K.X.); 2National Engineering Research Center for Non-Food Biorefinery, Guangxi Key Laboratory of Bio-refinery, Guangxi Academy of Sciences, 98 Daling Road, Nanning 530007, China; 3Guangdong Laboratory for Lingnan Modern Agriculture, South China Agricultural University, Guangzhou 510642, China

**Keywords:** microplastics, freshwater aquiculture, potential risk assessment, Pearl River, commercial species

## Abstract

The small size of microplastics and their wide distribution in water environments have attracted worldwide attention and heated discussion, because of their ingestion by aquatic organisms. At present, there are few studies on microplastics pollution in freshwater aquaculture ponds, especially shrimp ponds. In this study, the aquaculture ponds in the Pearl River Estuary were investigated. The abundance and composition of microplastics in different environmental media were studied to explore the potential sources and risk levels of microplastics, so as to provide basic data for the study of microplastics pollution in aquaculture ponds. Microplastics were observed in water and sediment samples at all sampling sites, with the abundance of 6.6 × 10^3^–263.6 × 10^3^ items/m^3^ (surface water) and 566.67–2500 items/kg (sediment), respectively. Thirty-seven individuals collected in six ponds belong to four species. Microplastics were observed in the gastrointestinal tract (GIT) of all fishes and shrimps, with the abundance ranging from 3–92 items/individual (fish) and 4–21 items/individual (shrimp). Among all samples, microplastics with the size range of <1 mm and fiber shape were the most common. The main microplastic components were cellulose, polypropylene (PP), and polyethylene (PE). The results of potential risk assessment showed that the pollution investigation of microplastics should not only consider the abundance. Low abundance does not mean low risk. Taking the toxicity score and abundance of microplastics as evaluation indexes to reflect the pollution status of microplastics may make the results more reliable.

## 1. Introduction

After weathering and wear, plastic waste is broken into small polymer particles with a length of less than 5 mm, which are called microplastics (MPs) [1,2,3,4]. According to their source, MPs can usually be divided into two categories: primary (they are intendedly produced in small size, mostly spherical and commonly used in personal care products) [5] and secondary MPs (formed by weathering of large plastic products, most of which are irregular in shape) [1]. Since the 1950s, the plastic industry in China has developed rapidly, and daily necessities made of plastics have been widely used [6]. MPs can be found in all types of environments, such as oceans [7], freshwater [8,9,10,11,12,13] and tap water [14]. About 300 million tons of microplastics enter the environment every year [15]. It is estimated that the plastic waste in 2015 exceeded 6 billion tons [16], of which only 1% was directly discharged into the marine environment, and most of them entered the soil and freshwater environment [17]. The different densities of microplastics lead to their different buoyancy, which makes them ubiquitous in water, and they can be ingested by aquatic organisms at different depths. A previous study showed that the aging of MPs will lead to cracks and fragmentation on the surface, thus increasing the surface area and carbonyl content and promoting the release of lead chromate pigment [18]. Many studies have shown that MPs are efficient adsorbents for hydrophobic/hydrophilic organic pollutants [19,20]. Furthermore, MPs can adsorb heavy metals (mainly through electrostatic adsorption and surface complexation) and transfer along the food chain, which is an important carrier of heavy metals migration in various environmental media, thus posing a huge challenge to ecosystem security [21,22]. Although studies have been conducted on MPs contamination in aquaculture [23,24], there are few studies on MPs contamination in freshwater aquaculture, especially in culture ponds. The rapid economic growth and urban development in recent decades have made the Nansha District seriously polluted by effluent (mainly from industry, agriculture, and daily life) and waste. In addition to having the largest mariculture area [25], China is the world’s largest freshwater aquaculture country. In 2015, China’s aquaculture production reached 67 million tons [26]. Pond farming is the most common farming mode in China, and its output accounts for 70% of total freshwater farming output [27]. Most aquaculture farms are traditionally operated on a small scale, lacking effective management measures [28]. Aquaculture sites, especially ponds, are good reservoirs for MPs. This closed farming environment is easily polluted by MPs, and MPs are not easy to be washed into the sea [24]. In highly urbanized and industrialized areas, microplastic pollution is serious, which may decrease the quality and quantity of aquaculture products [29,30,31]. In this study, fish ponds and shrimp ponds in the Nansha District of Guangzhou were selected as the research sites. Our research aims to reveal the contamination among different samples in aquaculture ponds, and to access the risk level of microplastics pollution in each aquaculture pond.

## 2. Materials and Methods

### 2.1. Description of Sampling Sites

Nansha is located at the geographical geometric center of the Pearl River estuary (located in the south-central part of Guangdong Province, China, formed by the outflow of the Pearl River to the South China Sea) and the Guangdong–Hong Kong–Macao Greater Bay Area, where the Xijiang River, Beijiang River and Dongjiang River meet. Its fishery resources are abundant, the existing contiguous aquaculture area exceeds 18 million m^2^, and the standardized aquaculture area is over 70%, making it one of the largest contiguous aquaculture bases in the province. Since the late 1970s, the Pearl River Delta region has become the most populous and economically active region in China [32]. Some studies have shown that the main stream of the Pearl River has a high abundance of MPs [33], so the culture pond near the Pearl River Estuary may be seriously polluted by MPs.

Four fish ponds in Shisiyong Aquatic Center (S_1_, S_2_: *Oreochroms mossambcus*; S_3_, S_4_: *Micropterus salmoides*) and two shrimp ponds (S_5_: *Penaeus vannamei* and S_6_: *Macrobrachium rosenbergii*) in Seagull Island were selected as sampling points (Figure 1).

### 2.2. Sample Collection

Samples were collected in October 2019 and the sampling sequence between different samples was defined before sampling. Commercial species need to be collected before collecting water samples and soil, because the collection process may startle the organisms, which can cause biological sample collection difficulties. Fish and shrimp samples (18 fish and 19 shrimps) from culture ponds were collected by casting a net and then the individual was wrapped in aluminum foil, placed in dry ice and transported back to the laboratory for subsequent treatments [23]. The containers used for sampling were rinsed with ultrapure water in advance. A 5 L stainless steel water collector was used to collect surface water samples, and three duplicate samples were collected from each culture pond [33]. The water sample was filtered with a 48 μm steel sieve, then rinsed the residue with ultrapure water and transferred it to a 100 mL glass container for subsequent treatment [34]. About 5 cm of pond sediment on the bottom surface of each culture pond, that was, the sediment sample of this study, was collected with a metal shovel (three duplicate samples were collected from each culture pond) [24]. Sediment samples were put into aluminum foil bags, the sampling point information and sampling time were also marked. 

### 2.3. Sample Pretreatment

Fifteen mL of 30% hydrogen peroxide were added to the water sample to digest organic matter, and then the water sample was left to stand for 24 h in dark conditions [3,35]. After digestion, a 0.45 μm filter membrane was used to filter water samples to obtain all the suspected microplastic particles. A slight modification of the density flotation method described by Zhao et al. [31] was adopted to separate MPs from sediments in this study. Then, 50 g of dried sediment were placed in a beaker and mixed with 400 mL of zinc chloride solution (density = 1.60 g/mL), and three parallel samples were needed for each sampling site. The mixed solution was thoroughly stirred with a clean glass rod and then covered with tin foil for 2 h of precipitation. Then, the supernatant was transferred to another clean 500 mL beaker. This separation step was carried out three times to improve the recovery rate of MPs. Finally, 15 mL of 30% hydrogen peroxide solution (H_2_O_2_) were added to the supernatant to digest the organic matter. After standing for 24 h, the solution was filtered by a vacuum device, and all particles were collected on the filter membrane. According to the research of Maes et al. [36], Zinc chloride was recommended for density flotation because its high density allowed flotation of most MPs [36]. Quinn et al. found that the recovery rate of MPs increased as the solution density increased [37]. Compared with saturated sodium chloride and saturated sodium iodide, the recovery rate of saturated zinc chloride was relatively high.

Body weight (g) and body length (cm) of fish and shrimps should be measured separately before dissecting biological samples (Appendix A). The gastrointestinal tract of fish was separated with a scalpel, and the gastrointestinal tract of shrimp was taken out from the back of the shrimp. The gastrointestinal tract of the fish was separated with a scalpel, and the gastrointestinal tracts of shrimps were taken out from the back of the shrimps. Then, 10% potassium hydroxide was added to the glass bottle containing the gastrointestinal tract (GIT) of fish and shrimp to submerge the tissues [38,39], and then the glass bottle was placed in an oven at 60 °C for at least four days. The glass bottle was shaken once a day to ensure complete digestion of organic matter [24]. After digestion, we filtered the solution through the filter membrane.

### 2.4. Microplastic Observation

The abundance, shape, size and color of MPs in samples were determined by observing the filter membrane with a stereo microscope. Based on the morphological characteristics of MPs, fibers, fragments, films, and pellets are considered to be the four main forms of MPs [40] (Figure 2). The sizes of MPs in this study are measured by software S-EYE and classified as follows: <0.5 mm, 0.5–1 mm, 1–2 mm, 2–3 mm, 3–4 mm, and 4–5 mm. Visual observation was mainly based on the physical characteristics of MPs described by Zhao et al., which mentioned that MPs are particles that were not easily damaged by tweezers, had uniform color and specific shape, and did not contain tissue or cellular structure [31]. 

Chemical composition analysis must be carried out to verify the accuracy of visual observation. Raman spectroscopy (Thermo Fisher Scientific DXR2, 532 nm laser, Raman shift 50–3500 cm^−1^) was used to identify the chemical composition of suspicious particles. To ensure the reliability of the results, only those spectra that match the standard database by more than 60% are accepted (compared with OMIC polymer spectra library) [41] (Appendix A).

### 2.5. Risk Assessment of MPs in Ponds

The potential risk pollution index method was first proposed by Swedish scientist Hakanson (1980) according to the properties and environmental behavior characteristics of heavy metals, which was used to evaluate heavy metal pollution in soil or sediment [42]. The method considers the content, ecological effect, environmental effect and toxicology of pollutants in soil, and takes into account the regional differences of background values. In addition to the assessment of the risk of pollutants in a specific region, it can also reflect the comprehensive impact of various pollutants, which is one of the good means to evaluate the potential ecological risk of sediments [30]. Therefore, this study used this method to analyze and evaluate the microplastic pollution in the sediments of aquaculture ponds. The formulas used are as follows:C^i^_f_ = C^i^/C^i^_n_;(1)
(2)$${T^{i}}_{r}\;=\;\sum_{n=1}^{n}P_{n}{\times \;S}_{n};$$
P_n_ = C_n_/C^i^;(3)
E^i^_r_ = T^i^_r_ × C^i^_f_(4)

In the formula, C^i^_f_ represents the pollution coefficient of a certain kind of MPs, C^i^ represents the measured value of MPs concentration in sediments, C^i^_n_ is the reference value needed for calculation, represents the concentration of MPs in the uncontaminated sample; and T^i^_r_ is the toxicity coefficient, which is used to reflect the toxicity level of MPs and the sensitivity of water to MPs in this study. P_n_ represents the percentage of MPs polymer types collected at each sampling site. S_n_ is the risk score of polymers containing MPs particles obtained by Lithner et al. [43]; C_n_ is the measured value of a certain microplastic at a sampling point; E^i^_r_ is a risk index that can be used to evaluate the potential risk level of a pollutant. 

### 2.6. Quality Assurance and Control

Blank experiments must be conducted to assess the potential contamination of the environment. All the solutions used in the experiment, including ultra-pure water, 30% H_2_O_2_, 10% KOH and ZnCl_2_ solution, were filtered according to the operation steps mentioned above, and three replicate samples were prepared for each solution. To reduce the MPs pollution from the laboratory, the experimenters should wear cotton lab clothes during the experiment [41]. All containers and anatomical tools must be cleaned with ultra-pure water prior to the experiment, and open containers needed to be covered with foil during microplastic extraction. 

### 2.7. Data Analysis

Data analysis is carried out on the SPSS (IBM, Chicago, IL, USA). To compare the abundance of microplastics in water and sediments in different regions, one-way ANOVA was applied to the analysis. Pearson test was used to determine whether there is a correlation between the abundance of MPs in water and sediment. Descriptive data such as maximum, minimum, mean and standard deviation were also used in this study.

## 3. Results

### 3.1. MPs in Surface Water

MPs were observed in all culture ponds. The abundance of MPs in surface water varied greatly (ranging from 6.6 × 10^3^–263.6 × 10^3^ items/m^3^), in which the abundance of S1 and S_2_ MPs was relatively low (Figure 3A). By contrast, the abundance of S_3_–S_6_ MPs was much larger than S_1_ and S_2_. The abundance of MPs was lowest in S_1_ and highest in S_3_. 

Given the results of component analysis, different types of polymers were identified in the selected samples, mainly cellulose, polystyrene (PS), polyethylene terephthalate (PET), and polyethylene (PE) (Figure 4). Consistent with other findings on seawater [44,45], fresh water [46,47], and mariculture [23,24], more small-sized MPs were found in surface water, especially <0.5 mm, accounting for 42.78% of all MPs in water samples (Figure 5C). The colors of MPs observed in surface water samples were mainly transparent (90.65%) and blue (7.62%) (Figure 5E). The dominant form of MPs was fibers, accounting for 98.04%, followed by fragments (1.14%) (Figure 5A). There was no significant difference in microplastic morphology among sampling sites.

### 3.2. MPs in Sediment

The abundance of MPs among sampling sites in sediments was ranging from 566.67 items/kg to 2500 items/kg, with an average value of 1330 ± 554.33 items/kg (Figure 3B). All sediment samples in culture ponds contained MPs, with the lowest values at S_1_, while highest in S_5_. Several kinds of MPs were found in sediment samples. After Raman identification, the main polymer types were cellulose, high-density PP, polymethylmethacrylate (PMMA) and PE (Figure 4). Fibers (81.73%) were the most common MPs morphology in sediments, followed by fragments (13.08%) and films (4.36%) (Figure 5B), which was consistent with the findings of Wu et al. [24]. By classifying and counting the sizes of MPs, the results showed that the number of MPs with small size was the largest (Figure 5D). As with surface water samples, MPs of various colors were observed in sediments, but the main colors were transparent (62.73%), blue (16.92%), pink (4.11%), green (3.65%), white (3.44%), yellow (3.40%) and red (3.40%) (Figure 5F). Pearson analysis results showed that there was no correlation between MPs in water samples and sediment samples.

### 3.3. MPs in Commercial Species

In this study, 37 biological samples of two kinds of fish and two kinds of shrimp were collected from six ponds, and MPs were detected in all biological samples. Among them, the abundance of MPs in *Oreochroms mossambcus* ranged from 3 to 13 items/individual, with an average value of 6.14 ± 3.80 items/individual; the microplastic abundance ranged from 12 to 92 items/individual in *Micropterus salmoides*, with an average value of 39.64 ± 23.38 items/individual; the abundance of MPs in *Penaeus vannamei* ranged from 4 to 21 items/individual, with an average value of 10.87 ± 4.94 items/individual; and the abundance of MPs in *Macrobrachium rosenbergii* ranged from 5 to 12 items/individual, with an average value of 9 ± 3.16 items/individual (Figure 6A). Since the gastrointestinal tissues were not weighed separately in this study, the MP abundance per gram of GIT was not calculated here. The species, weight and body length information of fish and shrimp are shown in Appendix A. Cellulose, PP and PE were the main polymer types in organisms (Figure 4). As for colors, transparent and blue dominated, while more yellow and white MPs were also observed (Figure 6C). The most abundant MPs were fibers, and a certain number of fragments were also found in the GITs of fish (Figure 6B).

### 3.4. Potential Risk Assessment of MPs in Sediments

The types of polymers observed in sediments at each sampling point in this study are shown in Table 1. Through the potential risk assessment of MPs in sediments, it was found that the abundance in sediment was lower than that of surface water, but the chemical toxicity of some detected polymers, such as PMMA and Polyvinyl chloride (PVC), cannot be ignored. Vinyl chloride, the main monomer of PVC, has been proved to cause hepatic angiosarcoma [48] and increase cardiovascular risk [49]. In addition, animals that inhaled vinyl chloride showed renal parenchymal lesions and swelling [50]. MPs in sediments can not only reflect the long-term interaction between land and water but also master the source and migration information of MPs [51,52,53]. Several studies have shown that, apart from density, the distribution of MPs is affected by other factors [54,55], especially in closed environments such as ponds, where the hydrodynamic is weak, the fluidity of MPs is poor and MPs may eventually tend to deposit in sediments. During the plastic manufacturing process, the polymerization reaction is incomplete, which leads to the release of unreacted residual monomers into the environment [56], posing a threat to the health of organisms and human beings. Raman results showed that some of the particles to be tested could not be identified as specific polymers, instead, they were identified as plastic additives, such as octadecanoic acid (commonly used as plasticizer and stabilizer) and copper (II) Phthalocyanine (commonly used in plastic spraying or dye). Combined with visual observation results, MPs have uneven structure, pits with different sizes on the surface and curled edges, indicating that MPs are obviously aged and degraded after weathering, thus residual monomers and additives added in the production process will be released into the environment [57]. In this study, the abundance of MPs and the chemical toxicity (toxicity of all monomers of polymer) of MPs were taken as indicators of risk assessment, and the chemical toxicity is represented by the risk score obtained by Litnner et al. [43]. Despite the high abundance of MPs, it was undeniable that more than half of the particles were identified as cellulose (mainly came from clothing), which was assumed to be non-toxic. Combined with the method of Peng et al. [30], E^i^_r_ × C^i^_n_ is used to compare the risks of each sampling site for the convenience of calculation because the value of C^i^_n_ is unknown (Table 2). The results of risk assessment showed that the aquaculture ponds in this study generally showed high risk, two high-hazard compounds and five relatively friendly compounds were found. Compared with S_3_–S_4_, the surface water of S_1_ and S_2_ showed lower MPs abundance, but there was no obvious difference in MPs abundance among the four sampling sites in sediment samples, and high-risk MPs were detected in S_1_ and S_2_. There are some limitations in using this method to carry out the pollution risk of MPs, because the background value of MPs is unknown, and the exact risk index value cannot be obtained. In addition, there is no clear basis for dividing the pollution risk level of MPs. In this study, the differences of risk levels in different ponds were generally reflected by this method, which indicated that the risk assessment of MPs could not only consider abundance data or observe MPs in water samples.

## 4. Discussion

### 4.1. Pollution of MPs in Surface Water

This study has opened up a new field for investigating the pollution of MPs in different media of pond water environment and evaluating the pollution risk of MPs in aquaculture ponds. Compared with other estuaries [34,58,59], freshwater fish ponds in the Carpathian Basin of Europe [60] and marine aquaculture areas in Xiangshan Bay [61], the abundance of MPs in surface water in this study is significantly higher (Appendix A). A study had been conducted to investigate the abundance of MPs in fish ponds at the Pearl River Estuary [62], whose abundance of MPs at the inlet of the fish pond was 32.1 items/L. Similarly, the Pearl River Basin in Guangzhou showed high MPs abundance (19.8 items/L) [63], from which we can draw a conclusion that the abundance of MPs in aquaculture water is obviously higher than that in pond influent. S_1_ and S_2_ had transferred fish before sampling, and then the two aquaculture ponds were idle, with only a small number of small fish in the pond. Therefore, compared with S_3_–S_6_, the lower abundance of MPs in S_1_ and S_2_ indicates that aquaculture activities are the main cause of high MPs abundance in fish ponds, such as feeding, fishing and degradation of feed woven bags. The woven bags filled with feed can also cause microplastics pollution to a certain extent, so further improvement should be made on feed packaging. Moreover, the fish pond is a closed environment with poor water exchange capacity and weaker hydrodynamic, which is a good reservoir for MPs [64]. In view of the morphology of MPs, except for the films (ANOVA, *P* > 0.05), there were significant differences in the number of fibers, fragments and pellets at each sampling site (ANOVA, *P* < 0.05). There was a great difference in microplastic abundance among repeated samples, which may be because the wind, water waves and fish swimming can have great disturbance on microplastic distribution in surface water in an enclosed environment. Therefore, the sampling sites in this study were separated by three meters to avoid the influence of exogenous factors on microplastic abundance.

### 4.2. Pollution of MPs in Sediment

Compared with previous studies [65,66], the abundance of MPs in sediments in this study is significantly higher (Appendix A). The results showed that fiber was the most dominant MP, followed by fragment and film; however, compared with surface water, the sediment contained more fragments, their smaller specific surface area made them easier to sink and accumulate at the bottom [67]. The low detection rate of pellets indicated that most MPs were derived from secondary MPs. As with surface water, sediments contained more small-sized MPs, which was consistent with the results of Di [47]. The high content of small-sized MPs may be due to mechanical or weathering degradation of large particles or direct crushing of plastic products [68,69]. The main color of MPs in sediments was transparent, and most MPs of this color came from fishing tools, such as fishing ropes and fishing nets [67,70]. However, the sediment contained a certain proportion of yellow MPs, which were highly degraded, indicating that the plastic degradation activity in the sediment of the culture pond was obvious, a large part of MPs come from far away and accumulate in ponds. There is no correlation between MPs in water samples and sediment samples, which could be explained by several factors, such as organism swimming, hydraulic action, MPs characteristics and so on [71]. 

### 4.3. Pollution of MPs in Commercial Species

In this study, the abundance of MPs in biological samples of different sizes and varieties was detected. The abundance of MPs observed in fish was higher than that in many previous studies [24,72,73,74,75,76] (Appendix A), and the GITs of shrimps in this study showed higher MP abundance than in the findings of Wu et al. [24], which may be related to the living environment and the individual size of shrimps. In general, shrimp is selective feeding [77], which enables it to refuse to eat non-edible substances, so the accumulation potential for MPs is relatively low. Compared with small Oreochroms mossambcus, Micropterus salmoides had a considerable proportion of large-size MPs and fragments in their GITs (Figure 6D), which may be related to the physiological characteristics of Micropterus salmoides. First, the Micropterus salmoides in this study were about one-year-old and larger; second, their big mouth enabled them easily intake, moreover, they were demersal organisms, and the high concentration of fragments at the bottom of the pond leads to high levels of fragments in their bodies, which was different from the results of Neves et al., indicating that habitat can affect MPs intake.

Furthermore, many yellow and white MPs were found in biological samples, and the yellow color was caused by the yellowing effect of MPs after long-term weathering in the environment. As some organisms are visual predators, it may affect the absorption of MPs by organisms [78,79]; most of the white MPs were identified as PP, and there were two main sources: 1, Fishing tools such as fishing nets; 2, The feed woven bag can be broken directly and enter the pond with the feed. By observing the abundance of MPs in the gastrointestinal tract of each species, it was found that the abundance of MPs in larger individuals (higher body weight) was higher. Therefore, the uptake of MPs by organisms is dependent on multiple factors, such as physiological characteristics, age, habitats and MPs pollution in the living environment.

### 4.4. Potential Effects of MPs on Freshwater Culture, Especially Pond Culture

Although the risk value of S_5_ and S_6_ was the lowest, the concentration of MPs was the highest among all sampling sites, and a certain amount of polyethylene terephthalate (PET) was detected in S_5_. Peng et al. thought that PET can often be considered as polyester when it appears as fiber [30], and polyester was considered to have a high risk score. Therefore, it is equally important to count the abundance of MPs and evaluate the chemical toxicity of different polymers. The lower detection rate of high-risk MPs in S_5_ and S_6_ indicated that shrimps have higher requirements on water quality, thus, the water purification equipment and measures in aquaculture ponds may play an important role in reducing the entry of exotic MPs into ponds. It is far from enough to detect the abundance of MPs in surface water only. The composition identification and risk assessment of MPs in sediments can provide a basis for the formulation of environmental policies and the management of aquaculture ponds.

## 5. Conclusions

MPs are an emerging pollutant with a small size and widely distributed in the water environment, which can easily be eaten by aquatic organisms and affects the quality and quantity of aquatic organisms. Aquaculture has played an important role in providing high-quality aquatic products for human beings. However, there are few studies on microplastics pollution in freshwater ponds. By studying the aquaculture ponds in the Pearl River Estuary, it was found that the aquaculture activities would cause microplastic pollution in the culture ponds. In addition, the abundance of MPs in the culture ponds was significantly higher than that in the main stream of the Pearl River and the pond inlet, indicating that the low flow rate and poor water exchange capacity in the ponds would cause the accumulation of MPs. Yellow MPs are highly degradable, so the exposure risk of harmful monomers is greatly increased. However, aquatic organisms easily ingest such MPs, and the negative effects on organisms need to be studied. The potential risk assessment of MPs in sediments showed that the hazard score and abundance of MPs were two important indicators for evaluating MPs pollution. Low abundance was not equal to low risk. High MPs abundance existed in shrimp ponds, but high-risk MPs were not observed. The water purification equipment in shrimp ponds may play an important role in reducing the pollution of exotic MPs. In the future, the risk levels of MPs in sediments should be classified to clarify the pollution status of MPs in the region.

## Figures and Tables

**Figure 1 ijerph-18-01869-f001:**
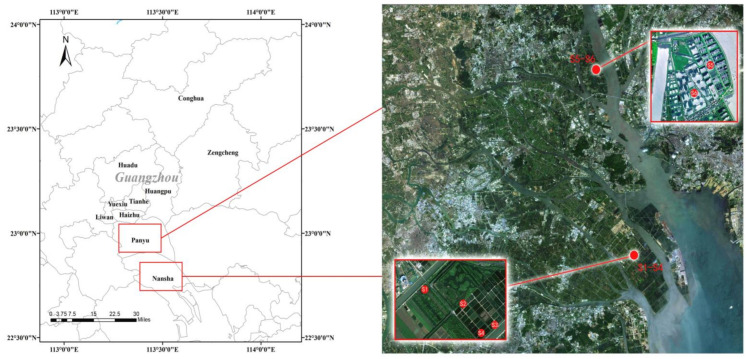
Geographical location of six sampling sites.

**Figure 2 ijerph-18-01869-f002:**
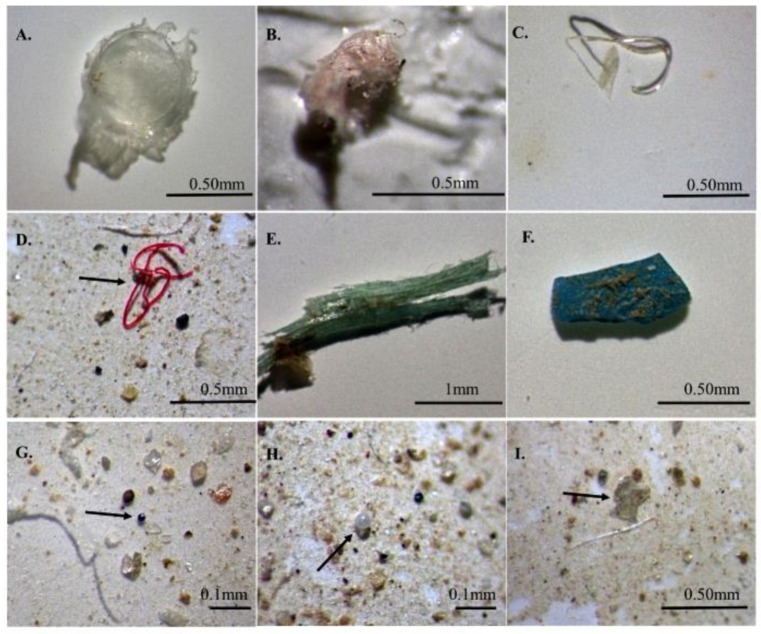
Photographs of different types of typical microplastics (MPs) observed in six aquaculture ponds (**A**–**I**): (**A**,**B**,**I**) films; (**C**,**D**) fibers; (**E**,**F**) fragments; (**G**,**H**) pellets.

**Figure 3 ijerph-18-01869-f003:**
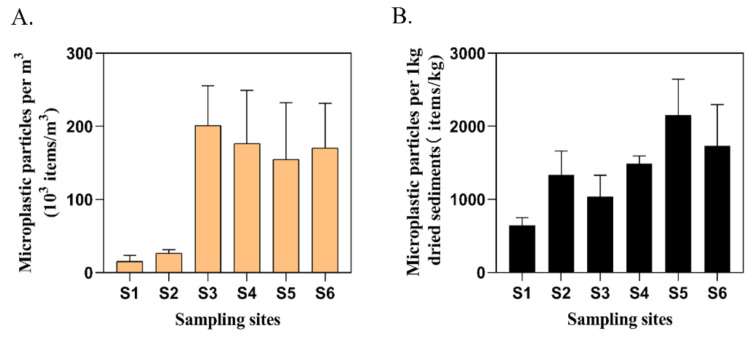
The abundance of MPs in surface water (**A**) and sediment (**B**) of six aquaculture ponds in the Pearl River Estuary (mean ± S.D.).

**Figure 4 ijerph-18-01869-f004:**
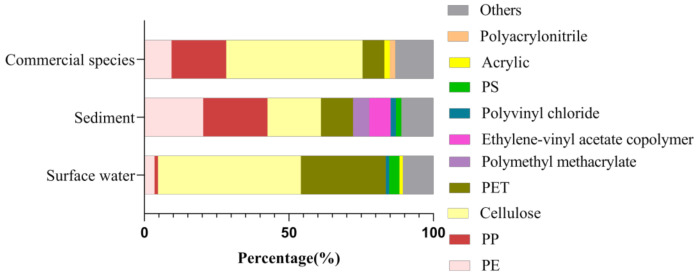
MPs composition in different samples of aquaculture ponds in the Pearl River Estuary.

**Figure 5 ijerph-18-01869-f005:**
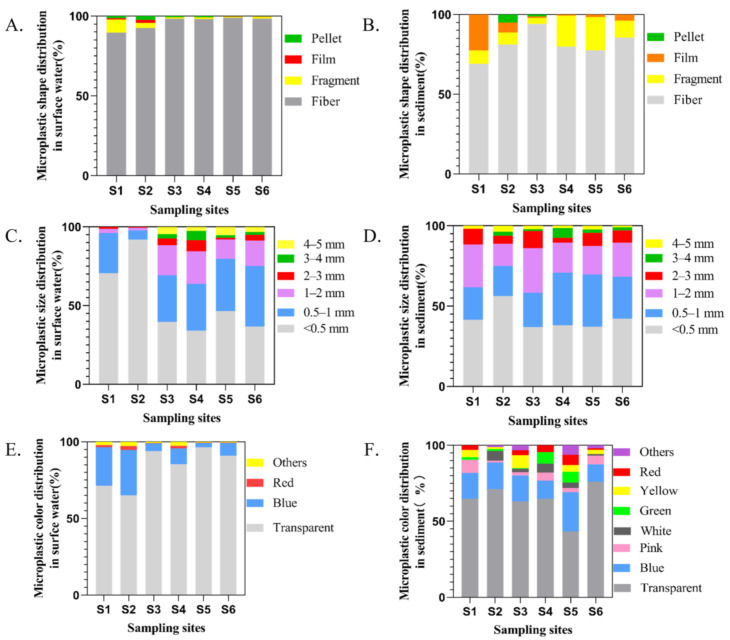
Types of microplastics in water (**A**) and sediments (**B**); size distribution of microplastics in water (**C**) and sediment (**D**); the proportion of microplastic color in water (**E**) and sediment (**F**).

**Figure 6 ijerph-18-01869-f006:**
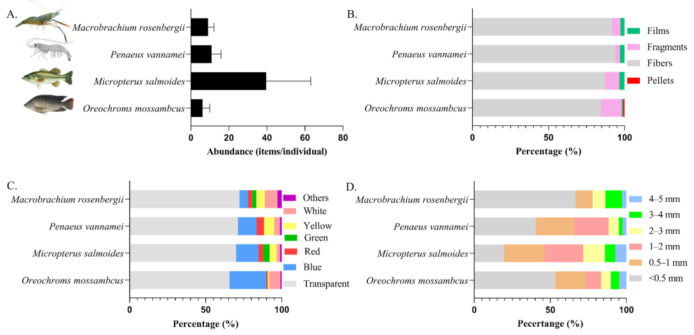
Microplastic abundance in gastrointestinal tract (GIT) of different commercial species in the Pearl River Estuary (mean S.D., n = 4–15) (**A**); types (**B**), color distribution (**C**) and size distribution (**D**) of microplastics in gastrointestinal tract of different commercial species in the Pearl River Estuary.

**Table 1 ijerph-18-01869-t001:** Identification results of microplastics in sediments at each sampling site.

Site	PP	PE	Cellulose	PS	PMMA	PET	Ethylene-vinyl Acetate Copolymer (EVA)	PVC
S_1_	3	4	2	0	1	0	1	0
S_2_	2	3	1	0	0	1	0	1
S_3_	2	0	2	1	1	1	0	0
S_4_	3	1	2	0	1	1	1	0
S_5_	1	1	1	0	0	2	1	0
S_6_	1	2	2	0	0	1	1	0
Total	12	11	10	1	3	6	4	1
Percentage	25%	22.92%	20.83%	2.08%	6.25%	12.5%	8.33%	2.09%

**Table 2 ijerph-18-01869-t002:** Potential risk assessment of microplastics in sediments of aquaculture ponds.

Site	Main Polymer	Hazard Level ^a^	The Value of S_n_ ^b^	The Value of E^i^_r_ × C^i^_n_ ^c^	Risk Level Estimation
S_1_	PP	I	1	1077	
	PE	II	11		Unfriendly
	PMMA	IV	1021		
	EVA	II	9		
S_2_	PP	I	1	10,590	
	PE	II	11		Unfriendly
	PVC	V	10,551		
	PET	II	4		
S_3_	PP	I	1	1057	
	PS	II	30		Unfriendly
	PMMA	IV	1021		
	PET	II	4		
S_4_	PP	I	1	1048	
	PE	II	11		
	PMMA	IV	1021		Unfriendly
	PET	II	4		
	EVA	II	9		
S_5_	PP	I	1	29	
	PE	II	11		Friendly
	PET	II	4		
	EVA	II	9		
S_6_	PP	I	1	36	
	PE	II	11		Friendly
	EVA	II	9		
	PET	II	4		

^a, b^ The hazard level of polymer and the value of S_n_ were obtained from [43]. ^c^ To facilitate the calculation, E^i^_r_ × C^i^_n_ is used to compare the risks between different sampling sites.

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
