# Peer review of "Microplastics Environmental Effect and Risk Assessment on the Aquaculture Systems from South China"

_ijerph, 2021, doi:10.3390/ijerph18041869_

Round 1
Reviewer 1 Report
The manuscript presents an interesting investigation, which covers microplastic (MP) detection in water, sediments and organisms. The outcomes are important and might contribute with the understanding on MP contamination and risk.
There are few minor points I'd suggest authors to address:
1- Abstract. Review 6.6x10^3;
2- please, review formulas presentation and variables indexes (e.g. line 158;
3- Please, specify what fluctuations bars mean on Fig 3;
4- Was ANOVA performed for the entire group of results, or for subsets? Please, clarify.
5- Fig 5 (c to F). Pleasae, consider to keep the same colour for the legends (e.g. same size range = same colour).
6- Why was not statistic significance tests performed, or presented, for MPs in commercial species? Please, clarify.
7- Risk level analysis should be better developed and explained. For example, what exactly means (Un)Friendly? I see calculations have been performed but the score seems much qualitative. Clarify, please.
Author Response
Reviewer #1:
* Abstract. Review 6.6x10^3.
Response: Thank you for your suggestion. We have collected the mistake.
* Line 158: please, review formulas presentation and variables indexes (e.g. line 158;.
Response: Thank you for your suggestion. We have collected the mistake.
* Line : Please, specify what fluctuations bars mean on Fig 3?
Response: Thank you for your question. The bar of this figure represents the mean value of microplastic abundance in water samples and sediments at each sampling sites.
* Was ANOVA performed for the entire group of results, or for subsets? Please, clarify
Response: Thank you for your question. The ANOVA was performed for the entire
group of results.
* Fig 5 (c to F). Pleasae, consider to keep the same colour for the legends (e.g. same size range = same colour).
Response: Thank you for your useful suggestion. I have change the color of the legends, please see line 233.
*Why was not statistic significance tests performed, or presented, for MPs in commercial species? Please, clarify.
Response: Thank you for your question. There are several kinds of biological samples, and the individual sizes of the same biological samples are different, so this study did not compare the biological samples between groups and within groups.
*Risk level analysis should be better developed and explained. For example, what exactly means (Un)Friendly? I see calculations have been performed but the score seems much qualitative. Clarify, please.
Response: Thank you for your useful suggestion. Different from the risk assessment analysis of microplastics in water samples, there is no specific method and evaluation standard for the risk assessment of microplastics in sediments at present, so this score can only reflect whether this compound is friendly in this environment (Peng et al., 2018).
Reference:
Peng, G., Zhu, B., Yang, D., Su, L., Shi, H., Li, D. Microplastics in sediments of the Changjiang estuary, China. Environ. Pollut. 2017, 225, 283-290.
Reviewer 2 Report
Review: ijerph-1089695
The manuscript reviewed investigated microplastics in aquaculture systems in the Pearl River Estuary, China. Water, sediment and biota (fish and shrimp) were counted and categorized based on type color, size and polymer type. The samples were collected from 4 fish ponds and 2 shrimp ponds in October 2019.
Although of interest to the global community, the small sample size and questions regarding statistical analysis weakens the scientific merit of the manuscript. These concerns needs to be addressed before the manuscript can be considered for publication.
Herewith some comments for the authors to address/consider:
- There are grammatical and typographical errors that needs to corrected:
- species names to be in italics (eg, see Line(L)224);
- reference to MPs and microplastics (suggest being consistent). See L65 and L66;
- sentence structure – eg L75-76; L127-128; L369-370;
- use of words eg L95 – is ‘operation’ the ideal word?
- check sentence tenses: eg L97-99; L123-128; L168-175;
- spelling: eg, L141 - ‘sprctra’
- check references: L265; 268
- Other comments:
- L69: I do not think sources of MPs can be determined from sampling the ponds alone. Perhaps remove/reword;
- Figure 1: text font too small. Suggest enlarging, inserting latitude/longitude and a scale bar;
- L75-78: suggest inserting a reference to substantiate;
- L80: if referring to ‘Some studies, suggest inserting references to substantiate;
- L83-85: sampling 6 ponds is a very small sample size for such a large area. Without a scale bar it is difficult to ascertain the distance between ponds. What criteria was used for selecting the ponds? Are the ponds representative of the total area (eg how many ponds are there in total for the area?). Random selection of ponds should have been considered and the authors are requested to indicate how the ponds selected are representative of all the ponds;
- L96-97 and L101: not sure what is meant by ‘three duplicate samples’. Suggest this be explained in more detail;
- L96 and L101: only 3 replicates were used per water and sediment samples, respectively. The low number of replicates reduces reliability of the values recorded and the subsequent statistical analyses is then questionable in my opinion, even if these were done in duplicate;
- L100-101: suggest providing more information on how sediment samples were collected in the middle of a pond with a shovel (without disturbing the sediment and potentially biasing sediment sampling); how deep were the ponds?
- L121: how was recovery rates determined?
- L123: suggest inserting units for weight and length measurements.
- L124: were the guts of the shrimps removed? Unlikely as these weighed 14-16 grams – very small; if so, explain how the guts were removed;
- L131: what magnification was used to record the MPs?
- L139: provide detailed information regarding the polymer analysis and identification (eg model of instrument, settings of the instrument and analyses done);
- L157: are the formulas correct? See formula (2) – appears to be incomplete;
- L157-166: check correctness of subscripts/superscripts; and the order in which it is explained – it does not follow the formulas as presented;
- L159-161: how was the value for this determined? Where was uncontaminated sample taken from?
- L168: how were blank measurements taken?
- L177-181: the statistical analysis is fundamentally flawed in my opinion. Was the data tested for normality and variances? Were the assumptions for parametric tests met? I doubt this to be the case given the low number of replicates. The authors must explain this section in more detail. If not, the stats in the results cannot be accepted with confidence.
- L185 & L 199: The concentrations reported here and elsewhere are very high. Are these correct? If these values are in fact correct, how was filtration with 45 um filter membranes possible? How was each MP particle recorded under the microscope (see L131);
- L199: which sites were significantly different? What post hoc test was done? What were the test values (F scores)
- Fig 3: the small and similar SD values seems unusual. Is this correct?
- L222: the small sample size of biota places the reliability of the results into question. Given the unequal sample size (Table S1), comparison within and between groups should be treated with caution;
- L230-231: this is the first reference to GIT being pooled. Is this for fish and shrimp? This must be explained in the M&M regarding how samples were processed and data analyzed;
- L275-276: what values were then used for the formulas presented in this study?
- L326: no correlations were reported in the results. Why report here?
- Conclusions should ideally not discuss new topics. Some of the concluding remarks could move to the discussion (use of woven bags)
- The discussion did not address potential sources of MPs. Either add to manuscript or remove from stated aim in L69.
Author Response
Reviewer #2:
*Line 224:species names to be in italics (eg, see Line(L)224).
Response: Thank you for your suggestion. I have corrected the mistakes, please see line 85-86.
*Line 65 and 66:reference to MPs and microplastics (suggest being consistent). See L65 and L66
Response: Thank you for your suggestion. As suggested, they have been corrected.
*sentence structure – eg L75-76; L127-128; L369-370;
Response: Thank you for your useful suggestion. I have used another presentation.
*use of words eg L95 – is ‘operation’ the ideal word?.
Response: Thank you for your suggestion. I have used another presentation, please see line 97.
*check sentence tenses: eg L97-99; L123-128; L168-175;
Response: Thank you for your suggestion. I have corrected the mistakes.
*spelling: eg, L141 - ‘sprctra’
Response: Thank you for your suggestion. I have corrected the mistake, please see line 151.
*check references: L265; 268.
Response: Thank you for your suggestion. I have added the references, please see line 284 and 285.
*Line 69: I do not think sources of MPs can be determined from sampling the ponds alone. Perhaps remove/reword
Response: Thank you for your suggestion. I have corrected the mistakes.
*Figure 1: text font too small. Suggest enlarging, inserting latitude/longitude and a scale bar
Response: Thank you for your useful suggestion. I have improved the figure.
*L75-78: suggest inserting a reference to substantiate;
Response: Thank you for your suggestion. There is no reference to this geographical description.
*L80: if referring to‘Some studies, suggest inserting references to substantiate
Response: Thank you for your useful suggestion. I have inserted the reference, please see line 83.
*L83-85: sampling 6 ponds is a very small sample size for such a large area. Without a scale bar it is difficult to ascertain the distance between ponds. What criteria was used for selecting the ponds? Are the ponds representative of the total area (eg how many ponds are there in total for the area?). Random selection of ponds should have been considered and the authors are requested to indicate how the ponds selected are representative of all the ponds.
Response: Thank you very much for your comments. In sampling areas, many same ponds were distributed in sampling sites. We only selected several representative for microplastic pollution studies. In addition, these ponds feed different fishes.
*L96-97 and L101: not sure what is meant by ‘three duplicate samples’. Suggest this be explained in more detail. only 3 replicates were used per water and sediment samples, respectively. The low number of replicates reduces reliability of the values recorded and the subsequent statistical analyses is then questionable in my opinion, even if these were done in duplicate.
Response: Thank you for your suggestion. In three repetitions, Three duplicate samples means three water samples and sediment samples were randomly collected from the same culture pond. Studies have been conducted to investigate the abundance of microplastics in surface water in other aquaculture ponds in the Pearl River Estuary. Its sampling method also follows threeduplicate samples principles.
References:
- Chen, M., Jin, M., Tao, P., Wang, Z., Xie, W., Yu, X., Wang, Assessment of microplastics derived from mariculture in Xiangshan Bay, China. Environ Pollut. 2018, 242:1146-1156. doi: 10.1016/j.envpol.2018.07.133.
- Zhu, J., Zhang, Q., Li, Y., Tan, S., Kang, Z., Yu, X., Lan, W., Cai, L., Wang, J., Shi, Microplastic pollution in the Maowei Sea, a typical mariculture bay of China. Sci Total Environ. 2019, 658:62-68. doi: 10.1016/j.scitotenv.2018.12.192.
- Ma, J., Niu, X., Zhang, D., Lu, L., Ye, X., Deng, W., Li, Y., Lin, Z. High levels of microplastic pollution in aquaculture water of fish ponds in the Pearl River Estuary of Guangzhou, China. Sci. Total Environ. 2020, 744, 140679.
*L100-101: suggest providing more information on how sediment samples were collected in the middle of a pond with a shovel (without disturbing the sediment and potentially biasing sediment sampling); how deep were the ponds?
Response: The pond is about 2 meters deep. With the help of fishing boats, the sampling points were randomly selected to take sediment samples. The handle of the shovel is about 1.5 meters long, and the shovel was put into the water to collect sediment samples from the bottom.
*L121: how was recovery rates determined?
Response:Thank you for your question. Quinn et al. found that the recovery rate of MPs increased as the solution density increased. Compared with saturated sodium chloride and saturated sodium iodide, the recovery rate of saturated zinc chloride was relatively high. In addition, multiple density flotation may increase the recovery rate of microplastics compared with single density flotation.
Reference:
Quinn, B., Murphy, F., Ewins, C. Validation of density separation for the rapid recovery of microplastics from sediment. Anal. Methods. 2017, 9, 1491–1498.
*L123: suggest inserting units for weight and length measurements
Response: Thank you for your useful suggestion. I have inserted the units, please see line 127.
*L124: were the guts of the shrimps removed? Unlikely as these weighed 14-16 grams – very small; if so, explain how the guts were removed.
Response: Thank you for your question. We do not have shrimp viscera treatment, but directly take the gastrointestinal tract of shrimps.
*L131: what magnification was used to record the MPs?
Response:The magnification usually adopted is 60. If we want to identify small size microplastics such as small balls, we will appropriately increase the magnification of eyepiece, and with the help of tweezers, the suspected particles of microplastics will be poked with tweezers under the microscope for identification.
*L139: provide detailed information regarding the polymer analysis and identification (eg model of instrument, settings of the instrument and analyses done)
Response:Thank you for your useful suggestion. I have added instrument model and settings, please see line 148.
*L157: are the formulas correct? See formula (2) – appears to be incomplete
Response:Thank you for your useful suggestion. I have collected my mistake, please see line 168.
*L157-166: check correctness of subscripts/superscripts; and the order in which it is explained – it does not follow the formulas as presented
Response:Thank you for your useful suggestion. I have collected my mistake, please see line 168-176.
*L159-161: how was the value for this determined? Where was uncontaminated sample taken from?
Response:The background value can not be determined, so the risk assessment value is represented by Eir × Cin in this study, and the Cin is just offset after multiplying the two. This calculation method is mentioned in Peng et al.
Reference:
Peng, G., Zhu, B., Yang, D., Su, L., Shi, H., Li, D. Microplastics in sediments of the Changjiang estuary, China. Environ. Pollut. 2017, 225, 283-290.
*L168: how were blank measurements taken?
Response:All the solutions used in the experiment, including ultra-pure water, 30% H2O2, 10% KOH and ZnCl2 solution, were filtered according to the operation steps mentioned above, and three replicate samples were prepared for each solution. In addition, during the experiment, a filter membrane was placed in the laboratory environment to observe the pollution of microplastics in the environment. Then the abundance of microplastics in the blank control was counted.
*L177-181: the statistical analysis is fundamentally flawed in my opinion. Was the data tested for normality and variances? Were the assumptions for parametric tests met? I doubt this to be the case given the low number of replicates. The authors must explain this section in more detail. If not, the stats in the results cannot be accepted with confidence.
Response:In this study, the fibers, fragments and pellets of surface water were analyzed by one-way variance. The data were tested by SPSS to see if they were in normal distribution. The sig. values were 0.06, 0.2 and 0.08, which were all greater than 0.05, and obeyed the normal distribution.
*L185 & L 199: The concentrations reported here and elsewhere are very high. Are these correct? If these values are in fact correct, how was filtration with 45 um filter membranes possible? How was each MP particle recorded under the microscope (see L131);
Response:Thank you for your question. We will explain from the following four aspects:
- At present, there are many literatures that use this sampling method. For example, Wang et al. pass the water sample through a 50 μm stainless steel sieve; Robin et al. used the trawl with a mesh size of 300 μm; Fok et al. filtered the supernatant with a 0.315 mm stainless steel sieve; Manta net used by Zhang et al. for sampling has a mesh size of 330 μm.
2.Most studies use different sampling methods, the lower limit of MPs detection is 0.3/0.33 mm (eg: Baldwin et al., 2016; Dris et al., 2015; Faure et al., 2015; Fischer et al., 2016; Leslie et al., 2017; McCormick et al., 2014; Su et al., 2016), and microplastics smaller than 0.3 mm are difficult to observe. Hidalgo-Ruz et al. mentioned that the mesh size used for sampling in previous studies was between 0.05 mm and 3 mm; Lusher et al. mentioned that the most commonly used mesh size is about 330 μm.
- A 0.45 μm filter membrane is used here, which is commonly used in laboratories, and it is used to collect particles in water samples, which can facilitate the observation and identification of microplastics.
- At present, the abundance of microplastics in freshwater culture area of Pearl River Estuary has been investigated. It can be found that the abundance of microplastics is on the high side and is almost in an order of magnitude in the breeding area.
Reference:
Wang, W.F., Ndungu, A.W., Li, Z., Wang, J., 2017. Microplastics pollution in inland freshwaters of China: a case study in urban surface waters of Wuhan, China. Sci. Total Environ. 575: 1369–1374.
Robin R S., Karthik R., Purvaja R., Ganguly D., Anandavelu I., Mugilarasan M., Ramesh R.(2020). Holistic assessment of microplastics in various coastal environmental matrices, southwest coast of India., 703, 134947. doi:10.1016/j.scitotenv.2019.134947
Fok, L., Cheung, P.K., Tang, G., Li, W.C., 2016. Size distribution of stranded small plastic debris on the coast of Guangdong, South China. Environ. Pollut. 1–6.
Zhang, W., Zhang, S., Wang, J., Wang, Y., Mu, J., Wang, P., Lin, X., Ma, D., 2017.
Microplastic pollution in the surface waters of the Bohai Sea, China. Environ. Pollut. 231, 541–548
Baldwin, A.K., Corsi, S.R., Mason, S.A., 2016. Plastic debris in 29 Great Lakes tributaries: relations to watershed attributes and hydrology. Environ. Sci. Technol. 50: 10377–10385. https://doi.org/10.1021/acs.est.6b02917.
Dris, R., Gasperi, J., Rocher, V., Saad, M., Renault, N., Tassin, B., 2015a. Microplastic contamination in an urban area: a case study in Greater Paris. Environ. Chem. 12:592. https://doi.org/10.1071/EN14167.
Faure, F., Demars, C., Wieser, O., Kunz, M., de Alencastro, L.F., 2015. Plastic pollution in Swiss surface waters: nature and concentrations, interaction with pollutants. Environ. Chem. 12:582. https://doi.org/10.1071/EN14218.
Fischer, E.K., Paglialonga, L., Czech, E., Tamminga, M., 2016. Microplastic pollution in lakes and lake shoreline sediments – a case study on Lake Bolsena and Lake Chiusi (central Italy). Environ. Pollut. 213:648–657. https://doi.org/10.1016/j.envpol.2016.03.012.
Leslie, H.A., Brandsma, S.H., van Velzen, M.J.M., Vethaak, A.D., 2017. Microplastics enroute: field measurements in the Dutch river delta and Amsterdam canals, wastewater treatment plants, North Sea sediments and biota. Environ. Int. 101:133–142. https://doi.org/10.1016/j.envint.2017.01.018.
McCormick, A., Hoellein, T.J., Mason, S.A., Schluep, J., Kelly, J.J., 2014. Microplastic is an abundant and distinct microbial habitat in an urban river. Environ. Sci. Technol. 48:11863–11871. https://doi.org/10.1021/es503610r.
Su, L., Xue, Y., Li, L., Yang, D., Kolandhasamy, P., Li, D., Shi, H., 2016. Microplastics in Taihu Lake, China. Environ. Pollut. 216:711–719. https://doi.org/10.1016/j. envpol.2016.06.036.
Ma, J., Niu, X., Zhang, D., Lu, L., Ye, X., Deng, W., Li, Y., Lin, Z. High levels of microplastic pollution in aquaculture water of fish ponds in the Pearl River Estuary of Guangzhou, China. Sci. Total Environ. 2020, 744, 140679.
*L199: which sites were significantly different? What post hoc test was done? What were the test values (F scores)
Response: Thank you for your suggestion. There is an error in my statement here, and
I have deleted this part of the statement, please see line 209.
*Fig 3: the small and similar SD values seems unusual. Is this correct?
Response:Thank you for your question. Actually, the SD is not small, because the
difference of ordinate in the graph is large.
*L222: the small sample size of biota places the reliability of the results into question. Given the unequal sample size (Table S1), comparison within and between groups should be treated with caution
Response:Thank you for your suggestion. Biological samples collected by Nie et al. also have different individual sizes. In this study, the abundance of microplastics in organisms in freshwater aquaculture ponds was investigated.
Reference:
Nie, H., Wang, J., Xu, K., Huang, Y., Yan, M. Microplastic pollution in water and fish samples around Nanxun Reef in Nansha Islands, South China Sea. Sci Total Environ. 2019, 696:134022. doi: 10.1016/j.scitotenv.2019.134022.
*L230-231: this is the first reference to GIT being pooled. Is this for fish and shrimp? This must be explained in the M&M regarding how samples were processed and data analyzed;
Response: Thank you for your question. ‘GIT” means the gastrointestinal tract of fish and shrimps. I have added the sample treatment method.
*L275-276: what values were then used for the formulas presented in this study?
Response: Thank you for your question. As a matter of fact, all but the background value (Cin) is unknown. The fourth formula is multiplied by the Cin, and the Cin is just offset after multiplying the two. So the Eir × Cin is used as the risk assessment value of this study. This calculation method is mentioned in Peng et al.
Reference:
Peng, G., Zhu, B., Yang, D., Su, L., Shi, H., Li, D. Microplastics in sediments of the Changjiang estuary, China. Environ. Pollut. 2017, 225, 283-290.
*L326: no correlations were reported in the results. Why report here?
Response: Thank you for your suuestion. I have collected my mistake, please see line 221.
*Conclusions should ideally not discuss new topics. Some of the concluding remarks could move to the discussion (use of woven bags)
Response:Thank you for your useful suggestion. I have collected my mistake, please see line 319.
*The discussion did not address potential sources of MPs. Either add to manuscript or remove from stated aim in L69.
Response: Thank you for your useful suggestion. I have collected my mistake, please see line 70.
Reviewer 3 Report
In this work the authors reported the study about different MP in fishes and shrimps. The manuscript turns out to be of great scientific interest, however, contains some deficiencies that need to be resolved before publication:
-Line 76: repeated phrase, please delete one and edit.
-Line 92: Insert the reference.
-Line 97: change “Filter the water sample” to “the sample water was filter”…..
-Line 139: Add “Raman spectroscopy was used”
-Line 141: change “sprctra” to “spectra”
-Line 216: The authors must improve the resolution of the figure.
Conclusions: authors must add the values found in the conclusion
Author Response
Reviewer #3:
*-Line 76: repeated phrase, please delete one and edit.
Response:Thank you for your useful suggestion. I have collected my mistake, please see line 75.
*-Line 92: Insert the reference.
Response: Thank you for your useful suggestion. I have inserted the reference, please see line 98.
*-Line 97: change “Filter the water sample” to “the sample water was filter”…..
Response: Thank you for your useful suggestion. I have collected the mistake, please see line 100.
*-Line 139: Add “Raman spectroscopy was used”
Response: Thank you for your suggestion. I have added, please see line 148.
*-Line 141: change “sprctra” to “spectra”
Response: Thank you for your useful suggestion. I have collected the mistake, please see line 151.
*-Line 216: The authors must improve the resolution of the figure
Response: Thank you for your suggestion. I have improved the resolution of figure, please line 224.
Reviewer 4 Report
The manuscript ijerph-1089695 entitled “Microplastics environmental effect and risk assessment on the aquaculture systems from South China” aimed to assess the potential sources and risk levels of microplastics in water, sediment fish and shrimps from aquaculture ponds located in the Pearl River Estuary (China).
The MS is interesting and provides new data to share with the scientific community. However, the section related to Material and Methods need to be better explored and explained. The M&M section should be described with sufficient detail to allow others to replicate and build on published results. For example, the section “sample collection” need to be clarified. How many fish/shrimps were collected from each pond? The authors state such information only in the results section. The keyworks need to be revised as they should not contain any words already in the title but can include abbreviated terms or location information not suitable for the title, since they will be used for indexing purposes. Furthermore, I recommend that you have your manuscript professionally edited by a native English speaker.
Other comments
Line 16: Please, specify the geographical location of Pearl River Estuary
Lines 83-85: Please, write the scientific name of fish/shrimp’s species in italics
Line 87: Please, add geographical coordinates of sampling sites (ponds)
Line 92: How many fish/shrimp samples did you collect? Please, specify
Line 99: Why did you select only the first 5 cm of sediment? Please, specify or add a reference(s).
Line 105: Please, replace “15” with “fifteen”
Line 109: Please, add [31] after Zhao et al., and remove it at the end of the sentence.
Line 118: Please, add a reference after Maes et al.
Line 123. How did you perform the fish/shrimp dissection to avoid contamination? Please, specify.
Line 177: Did your data normally distributed (i.e., Kolmogorov-Smirnov test)? Please, add a sentence.
Lines 222-223. Please, move the sentence to the M&M section.
Line 268: Please add a reference after Peng et al.
Author Response
*Line 16: Please, specify the geographical location of Pearl River Estuary
Response:Thank you for your suggestion. I have added the geographical location, please see line 76.
*Lines 83-85: Please, write the scientific name of fish/shrimp’s species in italics
Response: Thank you for your suggestion. I have collected my mistake, please line 85.
*Line 87: Please, add geographical coordinates of sampling sites (ponds)
Response: Thank you for your suggestion. I have added.
*Line 92: How many fish/shrimp samples did you collect? Please, specify
Response: Thank you for your question. We collected 18 fish and 19 shrimps respectively.
*Line 99: Why did you select only the first 5 cm of sediment? Please, specify or add a reference(s).
Response: Thank you for your question. I will explain it from the following two aspects.
- The thickness of sediment samples collected by the sampling tools used in this experiment is about 5 meters
- The acquisition depth can be referenced by references.
References:
Lin, L., Zuo, L., Peng, J., Cai, L., Fok, L., Yan, Y., L, H. Occurrence and distribution of microplastics in an urban river: A case study in the Pearl River along Guangzhou City, China. Sci. Total Environ. 2018, 644: 375–381, https://doi.org/10.1016/j. scitotenv.2018.06.327
Chen, M., Jin, M., Tao, P., Wang, Z., Xie, W., Yu, X., Wang, K. Assessment of microplastics derived from mariculture in Xiangshan Bay, China. Environ Pollut. 2018, 24:1146-1156. doi: 10.1016/j.envpol.2018.07.133.
*Line 105: Please, replace “15” with “fifteen”
Response: Thank you for your useful suggestion. I have collected the mistake, please see line 109.
*Line 109: Please, add [31] after Zhao et al., and remove it at the end of the sentence.
Response: Thank you for your useful suggestion. I have collected the mistake, please see line 113.
*Line 118: Please, add a reference after Maes et al.
Response: Thank you for your useful suggestion. I have added the reference, please see line 123.
*Line 123. How did you perform the fish/shrimp dissection to avoid contamination? Please, specify.
Response: First, the samples were immediately wrapped in foil and shipped back to the laboratory. Second, avoid using plastic when handling samples, the experimenters should wore cotton lab clothes during the experiment. Fish and shrimps were dissected on tin foil during the dissection.
Round 2
Reviewer 4 Report
The authors have addressed all my comments. However, the number of collected shrimps and fish must be added in the M&M section.
Furthermore, the scientific names of fish are not well written (i.e, line 236 and subsequent). They must be reported in italics
Author Response
The authors have addressed all my comments. However, the number of collected shrimps and fish must be added in the M&M section.
ANS:Shrimps and fish numbers have been added in the M&M section based your comments.
Furthermore, the scientific names of fish are not well written (i.e, line 236 and subsequent). They must be reported in italics
ANS:As suggested,it has been corrected.